# Analysis of Immune Associated Co-Expression Networks Reveals Immune-Related Long Non-Coding RNAs during MI in the Presence and Absence of HDC

**DOI:** 10.3390/ijms22147401

**Published:** 2021-07-09

**Authors:** Zhiwei Zhang, Suling Ding, Xiangdong Yang, Junbo Ge

**Affiliations:** 1Institutes of Biomedical Sciences, Fudan University, Shanghai 200032, China; 17111510047@fudan.edu.cn (Z.Z.); dingsuling1984@163.com (S.D.); 2NHC Key Laboratory of Viral Heart Diseases, Fudan University, Shanghai 200032, China; 3Key Laboratory of Viral Heart Diseases, Chinese Academy of Medical Sciences, Shanghai 200032, China

**Keywords:** myocardial infarction, histidine decarboxylase, immune infiltration, lncRNA

## Abstract

Myocardial infarction (MI) is one of the most common cardiovascular diseases. Although previous studies have shown that histidine decarboxylase (HDC), a histamine-synthesizing enzyme, is involved in the stress response and heart remodeling after MI, the mechanism underlying it remains unclear. In this study, using *Hdc*-deficient mice (*Hdc*^−/−^ mice), we established an acute myocardial infarction mouse model to explore the potential roles of *Hdc*/histamine in cardiac immune responses. Comprehensive analysis was performed on the transcriptomes of infarcted hearts. Differentially expressed gene (DEG) analysis identified 2126 DEGs in *Hdc*-deficient groups and 1013 in histamine-treated groups. Immune related pathways were enriched in Kyoto Encyclopedia of Genes and Genomes (KEGG) analysis. Then we used the ssGSEA algorithm to evaluate 22 kinds of infiltrated immunocytes, which indicated that myeloid cells and T memory/follicular helper cells were tightly regulated by *Hdc*/histamine post MI. The relationships of lncRNAs and the Gene Ontology (GO) functions of protein-coding RNAs and immunocytes were dissected in networks to unveil immune-associated lncRNAs and their roles in immune modulation after MI. Finally, we screened out and verified four lncRNAs, which were closely implicated in tuning the immune responses after MI, including ENSMUST00000191157, ENSMUST00000180693 (PTPRE-AS1), and ENSMUST-00000182785. Our study highlighted the HDC-regulated myeloid cells as a driving force contributing to the government of transmission from innate immunocytes to adaptive immunocytes in the progression of the injury response after MI. We identified the potential role of the *Hdc*/histamine-lncRNAs network in regulating cardiac immune responses, which may provide novel promising therapeutic targets for further promoting the treatment of ischemic heart disease.

## 1. Introduction

Myocardial infarction (MI) is one of the most lethal diseases with great morbidity [1]. It occurs when coronary arteries are blocked by ruptured plaques, which are constituted of lipids and leukocytes [2]. The sudden break of oxygen supplement leads to cardiomyocyte apoptosis [3] and the initiation of immune responses [3,4], including releasing cytokine/chemokine and the infiltration of different kinds of immunocytes [5]. However, the intrinsic linkage between ischemic injury and the immune responses remains unclear.

Histidine decarboxylase (HDC) is a key enzyme responsible for histamine production through conversing histidine to histamine [6]. Previous studies indicated that histamine [7] or the histamine receptor H3 agonist [8] can effectively protect heart failure or renal function damage caused by the loss of endogenous histamine in a model of cardiac dysfunction. Moreover, it was reported [9] that, after infarction, CD11b^+^HDC^+^ cells were recruited into the myocardium and involved in the inflammatory process and heart remodeling [10]. These data suggest that *Hdc* may participate in the immune response post MI through the regulation of immunocytes.

Long non-coding RNAs (lncRNAs) have been one of the hotspots in the field of RNA biology [11]. LncRNAs are defined as 200-nt long non-coding RNAs [12]. In contrast with short non-coding RNAs (miRNAs), lncRNAs are more multifunctional. They can act as sponges of miRNAs, decoys, scaffolds, guides, or enhancers, participating in the initiation of translation, the regulation of post-translation modulation, and the modulation of protein interactions [13]. However, lncRNAs involved in the regulation of the immune response in myocardial injury remain rarely reported. In this study, we constructed a model of myocardial infarction with *Hdc*-deficient mice [14] and performed RNA sequencing.

Differentially expressed genes (DEGs) were analyzed to explore the expression patterns and molecular functions. An immune infiltration algorithm was also used to quantify the ratio scores of different immunocytes recruited in injured myocardium. Finally, we constructed a correlation network among lncRNA, protein-coding mRNAs (pcRNAs), and enriched GO terms of pcRNAs to screen out the immune related lncRNAs (imm-lncRNAs) with functional prediction. Our work facilitates the study of immune reactions in cardiovascular diseases and provide novel targets for the immunomodulation of myocardial inflammation.

## 2. Results

### 2.1. Hdc-Deletion Induced Aggravated Infarction Injury in Hearts

We established a mouse model of myocardial infarction with wild-type (*Hdc*^+/+^) mice and *Hdc*-deficient (*Hdc*^−/−^) mice, which were treated with vehicle or histamine (HA) for 3 consecutive days before the operation (Figure 1A). To evaluate the efficacy of models, echocardiography analysis was performed in each group. Compared with mice in the control group, *Hdc*^−/−^ mice exhibited poorer heart function as indicated by a significantly lower ejection fraction (Figure 1B), an evidently higher end diastolic volume (Figure 1C) and end systolic volume (Figure 1D).

In addition, compared to Hdc^+/+^ mice, the percentages of fibrosis and collagen accumulation in the cardiac tissues of Hdc^−/−^ mice at 7 days post MI were more apparent as shown by the results of Masson’s trichrome staining (Figure 1E). The phenomena was effectively retarded by the administration of histamine, suggesting that *Hdc* deficiency greatly augmented the infarcted size post MI and that histamine treatment could rescue it.

### 2.2. Identification of DEGs at Different Time Point Post Myocardial Infarction in Hdc Deficiency Mice

To unveil the molecular mechanism underlying *Hdc* deficiency-aggravated myocardial infarction, we performed a time series of evaluations using RNA-sequencing. The RNA libraries were constructed from samples with or without active *Hdc* expression in triplicate at each time point. By analysis of the raw read counts, we used EdgeR [15] to identify differentially expressed genes (DEGs) at each time point. Their expressions were shown in the circus plot, according to the order of chromosomes (Figure 2A).

To confirm the genes that specifically changed responding to different conditions, we performed a differentially expressed analysis. Compared with the *Hdc*^+/+^ group, there were 2126 DEGs in the *Hdc*^−/−^ group, including 969 upregulated genes (374, 306, and 289 at day 0, 1, and 7) and 1157 downregulated genes (678, 271, and 208 at day 0, 1, and 7). Compared with the vehicle-treated group, there were 1013 DEGs in the histamine-treated group, including 498 upregulated genes (321, 45, and 132 at day 0, 1, and 7) and 515 downregulated genes (316, 137, and 62 at day 0, 1, and 7) (Figure 2B).

By comparing each group of DEGs, we identified the overlapping genes from different groups (Figure 2C), and their expression at different time points are also shown (Figure 2D). In comparison with the vehicle control, there were 15 overlapping genes changed simultaneously on the 1st and 7th day after MI in the histamine-treated group. Compared with the *Hdc*^+/+^ group, there were 132 overlapping genes changed simultaneously on the 1st and 7th day after MI in the *Hdc*^−/−^ group. Moreover, there were 12 overlapping genes appearing in the groups at day 1 post surgery and 45 overlapping gene in the groups with surgery at day 7.

### 2.3. Time-Dependent Changes in DEGs

To further describe the dynamic changes of gene expression profiles during the progression post MI, we stratified the DEGs of *Hdc* groups (*Hdc*^+/+^ vs. *Hdc*^−/−^) into six clusters according to the different trends and different degrees of the alteration of their levels. In the *Hdc* groups, genes of cluster 1 were only slightly downregulated at day 7; clusters 2, 3, and 4 represented that expression of gene decreased at day 1 and 7; the gene expression of cluster 5 and 6 at day 1 were decreased significantly, and only genes of cluster 6 were significantly increased at day 7 (Figure 3B).

Based on the same measure as above, the DEGs of histamine groups (Vehicle vs. histamine) were classified into four clusters (Figure 3A,C). The genes of all cluster 1 were only slightly downregulated at day 7; clusters 3 and 4 represented that expression of genes significantly decreased until at day 7; and the gene expression of cluster 2 at day 7 were increased dramatically.

The representative genes from each cluster were picked to show the pattern of gene expression at each time point (Figure 3B,D), which indicated that both *Hdc* deficiency and histamine treatment caused significantly different gene expression patterns in the heart with the progress post MI.

### 2.4. KEGG Functional Enrichment of DEGs

Kyoto Encyclopedia of Genes and Genomes (KEGG) analysis was then carried out to reveal enriched pathways altered across all time points [16] (Figure 4). Immune-related pathways were significantly and specifically enriched in the *Hdc*^−/−^ group versus the *Hdc*^+/+^ group at all the three time points, including antigen processing/presentation, Leishmaniasis, hematopoietic cell lineage, cytokine/cytokine receptor, and Rheumatoid arthritis. However, the pathways specifically enriched in the histamine-treated versus vehicle treated groups were less and scattered, and, apart from some immune-related pathways, NOD-receptor signaling and the protein digestion/absorption pathway were also enriched in these groups.

### 2.5. Immune Cell Profiles of Ischemic Heart in a Histamine-Treated/Hdc-Deficient Myocardial Infarction Model

To clarify the effects of *Hdc*/histamine on the immune cell infiltrated into the ischemic heart, we used the ssGSEA algorithm [17] to estimate the relative amounts of different immunocytes infiltrated post MI. The infiltration of neutrophils, Eosinophils, myeloid-derived suppressor cells (MDSCs), and T memory/follicular helper cells increased significantly in *Hdc*^−/−^ mice at day 1 and 7 after MI, which were significantly decreased with histamine treatment (Figure 5A). The infiltration of macrophages increased at day 1 but decreased at 7 days post MI in *Hdc*^+/+^ mice, while it continued to increase to the 7th day post MI in *Hdc*^−/−^ mice, which disappeared in histamine-treated *Hdc*^−/−^ mice (Figure 5A).

Interestingly, although multiple T cells and NK cells did not increase in the *Hdc*^−/−^ groups, they decreased responding to histamine treatment. Notably, the infiltration of B cells was induced in *Hdc*^−/−^ mice and decreased with histamine treatment (Figure 5A). Combining the previous report that a patient with B cell chronic lymphocytic leukemia (B-CLL) subsequently developed cutaneous infiltrates [18], our finding implies that *Hdc* may contribute to the development or functions of B cells in myocardial inflammatory response.

Then, we evaluated the correlation of the immunocytes to explore the potential relationships between them, by Person’s correlation analysis (Figure 5B). Among all the immunocytes, all subtypes of B cells (naïve/memory B cells and plasma cells) were strongly correlated with plasmacytoid dendritic cells (pDC); myeloid cells, including macrophages, neutrophils, and eosinophils, were strongly correlated with each other; mast cells, such as histamine secreting cells, were strongly correlated with basophils. However, basophils were negatively correlated with T memory cells; neutrophils were negatively correlated with pDCs and B cells; and mast cells were negatively correlated with eosinophils, macrophages, and monocytes.

To evaluate the effects of *Hdc*/histamine on the composition of infiltrated immunocytes, we accessed the proportion of every immunocyte across the samples. We found that cytotoxic T cells were the most discrete among the 22 cell types. The cell ratio analysis indicated that cytotoxic T cells were decreased with *Hdc* deficiency but could not be reversed with histamine treatment (Figure 5C,D).

### 2.6. Functional Co-Expression Network of Imm-lncRNAs

To screen out the immune-related lncRNAs (imm-lncRNAs), we first calculated the correlations between lncRNAs, pcRNAs, and immunocyte infiltration scores. Then, we used the lncRNAs with a high coefficient to construct the lncRNA-pcRNA (Figure 6A) and lncRNA–immunocyte networks (Figure 6B). We identified 15 hub genes corresponding to 19 infiltrated immunocytes.

Next, we enriched the biological functions of mRNAs, which were strongly correlated with the top six immune-related lncRNAs (coefficient > 0.75, *p* value < 0.001), by constructing a multilayer co-expression functional network (Figure 6C). The results showed that ENSMUST00000182785 was highly correlated with the T memory cells (coefficient > 0.81), potentially through influencing PDGF/heparin/glycosaminoglycan binding and Rac/Rho GTPase activity. ENSMUST00000180863, involved in cytokine receptor activity, was correlated with neutrophils (coefficient > 0.8). 

ENSMUST00000181565 and ENSMUST00000180748 were highly correlated with macrophages, monocytes, and dendritic cells (coefficient > 0.79 and 0.74 respectively), potentially through regulating microtubule/tubulin/motor binding, ATPase/GTPase activity, and histone kinase activity. ENSMUST00000180693 was co-related with macrophages, monocytes, and T memory cells (coefficient > 0.68), potentially through affecting enzyme activator activity and nucleosomal DNA binding. ENSMUST00000191157 was highly co-related with macrophages, monocytes, and eosinophils (coefficient > 0.78), likely participating in cytokine/chemokine/receptor signaling, the Rac guanyl-nucleotide exchange factor/GTPase pathway, and phospholipase/phosphatidylinositol phosphate binding.

### 2.7. qPCR Verification of Immune-Related lncRNA Expression in Hdc^−/−^ Mice Model of MI

Finally, we used qPCR to validate the levels of the lncRNAs selected above (Figure 7A–C). The results showed that ENSMUST00000182785 was down-regulated in the *Hdc*^−/−^ group versus the *Hdc*^+/+^ group both before and after MI surgery. However, the decreased range of the ENSMUST00000191157 level reached significance only in the sham groups, while the reduction of ENSMUST00000180693 (PTPRE-AS1) reached significance only in the MI groups. Histamine treatment partially elevated the PTPRE-AS1 level reduced by Hdc deficiency but did not reach a significant difference (Figure 7C). 

We speculated that this may be because PTPRE-AS1 is not specifically expressed in immune cells, the difference of its expression in immune cells regulated by Hdc/histamine had be concealed by its expression in other cells, such as cardiomyocyte and cardiac fibroblasts. Thus, we explored the effect of histamine on the expression of PTPRE-AS1 in vitro. As shown in Figure 7D, the level of PTPRE-AS1 could be evidently up-regulated by histamine treatment in vitro.

## 3. Discussion

Myocardial infarction is one of the leading causes of mortality [1]. In spite of the advancement of pharmacology and percutaneous coronary intervention (PCI), heart failure and arrhythmia are still inevitable in clinical practice [19,20]. Accordingly, this drives us to elucidate the comprehensive molecular regulatory network underlying its pathogenesis urgently.

Clinical trials, such as the Canakinumab Anti-inflammatory Thrombosis Outcomes Study (CANTOS) [21] and the Colchicine Cardiovascular Outcomes Trial (COLCOT) [22], have proven that patients with myocardial infarction could accrue benefit from anti-inflammation treatment. However, the investigation on the correlations of each kind of immunocyte during cardiovascular inflammation is still sparse due to the limited techniques [3]. Recently, the application of flow cytometry, immunofluorescence, and single cell sequencing has greatly facilitated our understanding of the diversity of immunocytes and their individual contribution to the reparation of ischemic injury [5].

*Hdc* has been implicated to take part in the inflammatory responses after the ischemic injury of hearts. Activation, rather than inhibition of histamine signaling, contributes to the recovery of MI or heart failure [8]. Through combining RNA sequencing and bioinformatics analysis, the results of this work suggested that *Hdc*/histamine signals participated in regulating the pattern and the correlation of the immunocytes infiltrating into heart post MI at least partly via affecting the levels of imm-lncRNAs.

Long non-coding RNAs (lncRNAs) have recently attracted huge attention due to their great potential involving various biological processes [12], while the roles of lncRNAs in the immune response post MI is still unclear. In this study, RNA-sequencing was carried out to evaluate the expression of mRNAs, including protein-coding RNAs and lncRNAs, in the ischemic hearts of *Hdc*-deficient mice with or without histamine treatment.

Compared with the *Hdc*^+/+^ control groups, we identified 1052, 577, and 497 DEGs in *Hdc*^−/−^ mice at 0, 1, and 7 days post MI, respectively; while, in *Hdc*^−/−^ mice pretreated with histamine, the numbers of DEGs reduced to 637, 182, and 194 at 0, 1, and 7 days post MI, respectively. There were less DEGs and, hence, less pathways enriched in the histamine-treated groups, including protein digestion/absorption and IgA production. The results indicated that *Hdc* deficiency may profoundly impair the reactions of the immune system responding to injury, and this could be partially rescued by pretreatment with exogenous histamine through retuning the gene expression network.

Based on DEGs, KEGG analysis was performed to demonstrate that different biological/disease pathways were enriched at specific time points following different treatments. For example, on day 1 and 7 after MI, immune-reaction-related pathways were activated, including hematopoietic cell lineage, cytokine-cytokine receptor, IL-17 signaling, and B-cell-receptor signaling. Whereas, other pathways were only activated on day 7, including NK-cell-mediated cytotoxicity, T-cell-receptor signaling, and NOD-like-receptor signaling.

The immune related DEGs enriched from day 1 to day 7 indicated a delayed or persistent immune response in *Hdc*^−/−^ mice post MI. However, the levels of T-cell-, NK-cell-, and B-cell-related genes were significantly altered on day 7 after MI, which is usually the time point when the transition from innate immunocytes to adaptive immunocytes occurs after MI. Thus, although it is suggested that *Hdc* is mainly expressed in myeloid cells of innate immunocytes in the circulation, our results implied that adaptive immunocytes may also be influenced by *Hdc* deficiency, either directly or indirectly.

In accordance with the KEGG analysis, the ssGSEA analysis also showed that the pattern immunocytes infiltrated into the heart were significantly altered by *Hdc*/histamine signaling. The amounts of myeloid cells, DC cells, and macrophages especially, were significantly higher in infarcted hearts of *Hdc*^−/−^ mice but decreased with histamine treatment. This may be due to the important roles of histamine signaling in myeloid cells. Though the number of NK cell and T cytotoxic cells did not show any differences between *Hdc*^+/+^ and *Hdc*^−/−^ mice, they decreased significantly responding to histamine treatment. It was previously reported that HDC^+^ granulocytic myeloid cells induced the proliferation of T regulatory cells via the Cxcl13/Cxcr5 axis [23]. Consistent with this, T regulatory cells emerged on day 1 but disappeared on day 7 post MI in *Hdc*^−/−^ mice, which was opposite to the trend in *Hdc*^+/+^ mice.

T follicular helper (Tfh) cells are antigen-presenting cells that are induced by CD4^+^ T cells and support B cell differentiation [24]. In our data, the infiltration scores of Tfh cells were up-regulated in the *Hdc*^−/−^ groups and down-regulated in the histamine-treated groups. Moreover, B cell subsets had a similar infiltration pattern with Tfh cells, and the Tfh cell was strongly correlated with these B cell subsets. These imply that the aberrant infiltration of B cells in *Hdc*^−/−^ groups may be driven by HDC^−/−^ antigen-presenting-cells, DC cells, or macrophages.

To investigate the potential functions of imm-lncRNAs, we constructed a lncRNAs-GO-immunocyte network by combination of the immunocyte-lncRNA correlation network and lncRNA-mRNA co-expression network, which included six lncRNAs, 20 GO pathways, and eight immunocytes. Among them, we verified lncRNAs, including ENSMUST00000180693 (PTPRE-AS1), ENSMUST00000191157, and ENSMUST00000-182785.

PTPRE-AS1 locates on the antisense strand of PTPRE and has recently been reported to regulate the polarization of macrophages [25]. PTPRE-AS1 deficiency enhanced PTPRE expression by regulating WDR5-dependent H3K4me3 and, hence, promoted IL-4-stimulated M2 macrophage polarization [25]. Reparative M2 macrophages have been found to be involved in tumor proliferation and fibrosis [26]. We found that the expression of PTPRE-AS1 significantly decreased in *Hdc*^−/−^ mice post MI and increased in RAW264.7 with histamine treatment, which implied that PTPRE-AS1 may play a role in monocyte/macrophage-dependent susceptibility to carcinogenesis [6] and heart fibrosis [10] in *Hdc*^−/−^ mice.

ENSMUST00000182785 locates in the intergenic regions harboring large numbers of non-coding RNAs, including lncRNA and miRNAs, implying that they may function as sponges of these ncRNAs. Based on the location in the genome, ENSMUST00000191157 may be involved in regulating the expression and function of Ipo7 or Zfp1, which are nuclear located.

Although the exact functions of these lncRNA are still poorly understood, our results provided important evidence and clues for further studies of their roles in the immune response of MI. This research not only unveiled potential pathways participating in *Hdc*-regulated immune responses but also provided hints for further exploring the mechanisms of immunomodulation of MI.

Inevitably, there were some limitations in our study. First, since *Hdc* is mainly expressed in myeloid cells that are recruited into the myocardium, immunocytes of the adaptive immune system may be less discussed in our study. Although the screened lncRNAs were observed to be significantly down-regulated in the *Hdc*^−/−^ group, we are not sure if the change is the reason or if it results from the alteration of the immune-infiltration pattern caused by *Hdc* deficiency. We identified a few potential mechanisms involved in the regulation of the immune response by lncRNAs; however, more experiments are needed to elucidate the detail roles of lncRNAs in the signaling pathways involved. In addition, the dynamic change of immunocyte recruitment could be more elaborately observed if the transcriptome data for more time points could be collected.

## 4. Materials and Methods

### 4.1. Animals and Establishment of the MI Mouse Model

*Hdc*-deficient (*Hdc*^−/−^, Balb/C background) mice were generously provided by Professor Timothy C. Wang from Columbia University. The generation of *Hdc*^−/−^ mice has been described in previous papers [14]. Male 8 to 12-week-old wild-type (*Hdc*^+/+^ in Balb/C background) mice were purchased from CAVENS lab animals. Recently, there was a newly published work suggested that 25.5 to 27.6 °C is the best temperature range for housing C57BL/6 mice to mimic human thermal relations [27]. However, due to the different strains, the mice in this work were still housed at 22 °C under a 12 h light/dark cycle as in the past, with free access to food and water. 

All procedures in this study conformed to the Guide for Animal Management Rules from the Ministry of Health of the People’s Republic of China, and approved by the Institutional Review and Ethics Board of Fudan University. The myocardial infarction model was performed as described before [28]. Briefly, male mice were anaesthetized with 3% isoflurane and a small incision was made at the fourth intercostal space to pop out the heart. The left main descending coronary artery was permanently ligated. After the heart was placed back into the intrathoracic space, fast air evacuation and chest wall closure were performed. Histamine (Sigma, MO, USA) was injected 4 mg/kg/d intraperitoneally beginning from 3 days before surgery.

### 4.2. Echocardiography

*Hdc*^+/+^ and *Hdc*^−/−^ mice were anaesthetized with a 1.5% isoflurane at day 7 after myocardial infarction. Two-dimensional short axis images were recorded and analyzed using the Vevo2100 System (Visual Sonics, Toronto, ON, Canada).

### 4.3. Histological Analysis

Hearts with operations were fixed 10% formalin and embedded in paraffin. We used 5 μm thick paraffin sections for Masson’s trichrome staining. Fibrosis was expressed as a percentage of the total ventricular area.

### 4.4. RNA-Seq and Data Processing

The total RNA was extracted from mice euthanized at the indicated time after surgery, using the TRIzol reagent (Invitrogen, CA, USA). RNA library construction used the RNA Library Prep Kit for Illumina^®^ (NEB, UK) following the manufacturer’s recommendations. Then, polymerase chain reaction (PCR) was performed for the ligated adaptors. The cDNA library concentration was measured using a Qubitfi 2.0 fluorometer; each sample was mixed to the same quality. The Illumina Hiseq 2500 platform was used to sequence the library, and RNA-seq reads were performed using the Bowtie [29] (v2.0.6)-TopHat [30] (v2.0.9) pipeline.

### 4.5. Transcriptome Analysis

The differentially expressed genes (DEGs) between each group were generated with R-package EdgeR (fold change ≤ −0.5 or ≥0.5; *p* < 0.05). Next, we used ggtern [31] to depict the expression of DEGs in different groups. Then, we used stem program to find clusters in time-series [32].

The DEGs were mapped to Kyoto Encyclopedia of Genes and Genomes (KEGG)/the Gene Ontology (GO) [33], by using R-package clusterProfiler [34]. Immune infiltration analysis was performed using the ssGSEA algorithm, and the infiltration scores of 22 types of immunocytes were calculated using the signature genes [35] from https://panglaodb.se/index.html (accessed on 10 January 2021). Pearson’s correlation coefficients were calculated to build the networks between immunocytes, lncRNAs, and protein-coding RNAs. Co-expression networks were constructed using Cytoscape [36] (v3.6.0).

### 4.6. Cell Culture

RAW264.7 cells were maintained in high-glucose DMEM medium (Gibco, Bleiswijk, The Netherlands) with 10% fetal bovine serum. Histamine (0.1 mM) or PBS was added into the culture medium for 24 h.

### 4.7. Quantitative Real-Time PCR

The cDNA synthesis was carried out using PrimeScriptTM kit (Takara, Dalian, China) after RNA isolation, according to the manufacturer’s instructions. Quantitative real-time PCR using TB GreenTM qPCR Mix (Takara, Dalian, China) was performed in a Bio-Rad CFX ConnectTM Real-Time Detection System. The relative expression levels were determined by normalizing to *Actb* (Table 1). The changes in the relative gene expression normalized to the internal control levels were determined using the relative threshold cycle method.

### 4.8. Statistical Analysis

The data are shown as the mean ± SEM. Student’s *t*-test was performed for two-group comparisons, and one way ANOVA analysis of variance was used for multiple comparisons. The value of *p* < 0.05 was considered statistically significant.

## 5. Conclusions

In conclusion, we comprehensively evaluated the immune cell landscape that infiltrated into the ischemic heart in the presence and absence of HDC through transcriptomic analysis. The results highlighted the HDC-regulated myeloid cells as a driving force contributing to the government of transmission from innate immunocytes to adaptive immunocytes in the progression of injury response after myocardial infarction. Through constructing the linkage between lncRNAs and cardiac immune responses, the study screened three novel imm-lncRNAs that potentially mediated histamine signaling in regulating the mobilization, recruitment, and differentiation of immunocytes post MI. We believe these findings will be valuable for developing new strategies to retune adverse immune responses after myocardial infarctions.

## Figures and Tables

**Figure 1 ijms-22-07401-f001:**
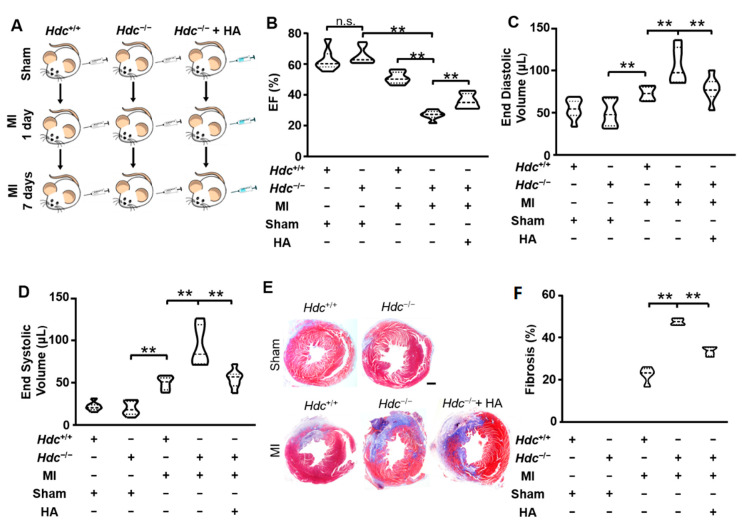
*Hdc* deletion reduced heart function at day 7 after MI. (**A**) Experimental design for transcriptomic analysis of *Hdc*^+/+^ and *Hdc*^−/−^ mice, with or without histamine (HA, 4 mg/kg) treatment, at 0, 1, or 7 days post myocardial infarction surgery. n = 3 per time point. (**B**–**D**) Quantitative analysis of left ventricle ejection fraction (**E**,**F**), end diastolic volume and end systolic volume of mice at the 7th day post MI surgery. n = 5. (**E**) Representative figures of Masson’s trichrome staining of cardiac mid-sections from *Hdc*^+/+^ and *Hdc*^−/−^ mice, with or without histamine. The myocardial fibers and collagen are colored blue. Quantitative analysis of the percentages of fibrosis is shown on the right. ** *p* < 0.01, n.s., no significance.

**Figure 2 ijms-22-07401-f002:**
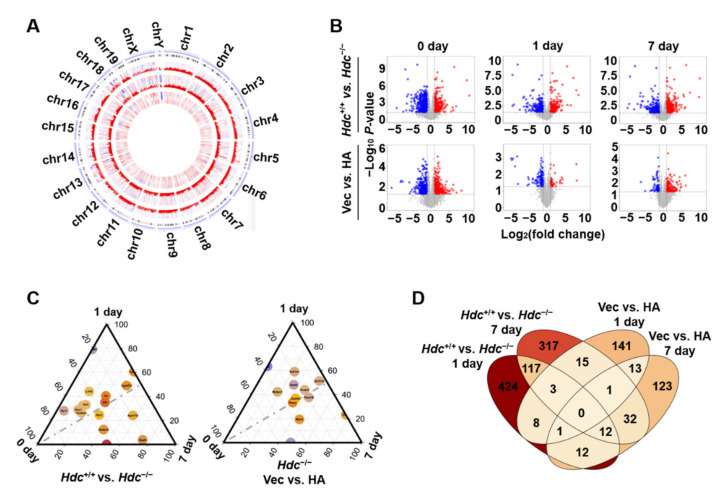
The gene expression profiling of the mice in each group. (**A**) Circos heatmap of differentially expressed genes. DEGs are present in the order of chromosomes. The bigger circles represent the DEGs at day 0, 1, and 7 after MI. The smaller circles compare the expressions of DEGs among *Hdc*^+/+^, *Hdc*^−/−^, and *Hdc*^−/−^ with histamine treatment. (**B**) Volcano plots display DEGs with *Hdc* deficiency or histamine treatment at day 0, 1, and 7 after MI. Transcripts up-or down-regulated are in red or blue and those unchanged are in gray. (**C**) Venn diagram of DEGs overlapping at different experimental time points and under different treatments. (**D**) Triangle pictures show the expression of DEGs at day 0, 1, and 7 after MI, comparing *Hdc*^+/+^ vs. *Hdc*^−/−^ (**left**), and Vec vs. HA (**right**).

**Figure 3 ijms-22-07401-f003:**
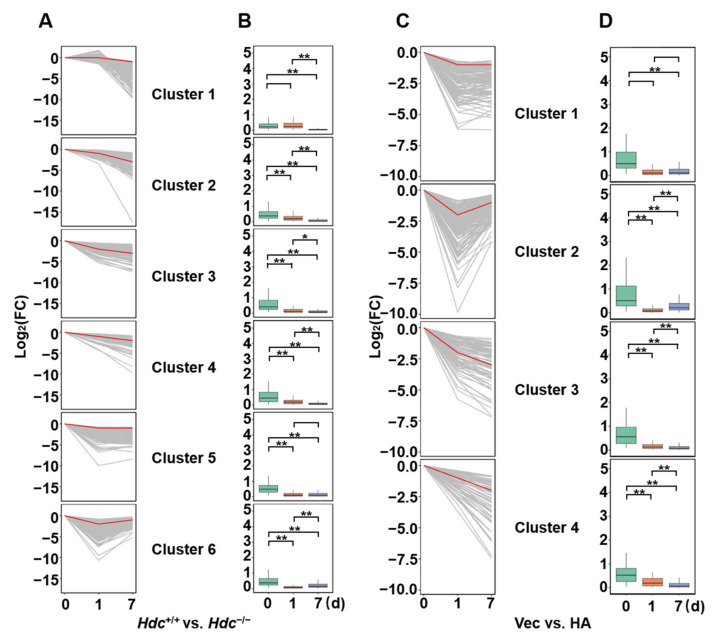
The diagrams depict the patterns of dynamic expression of DEGs. (**A**,**C**) Gray lines represent individual gene expressions, and red lines represent the summarized values for each pattern. (**B**,**D**) The diagrams show the change of representative genes during the progression of MI. The green, red and blue boxes represent the relative expression of genes at day 0, 1, and 7 after MI. * *p* < 0.05, ** *p* < 0.01.

**Figure 4 ijms-22-07401-f004:**
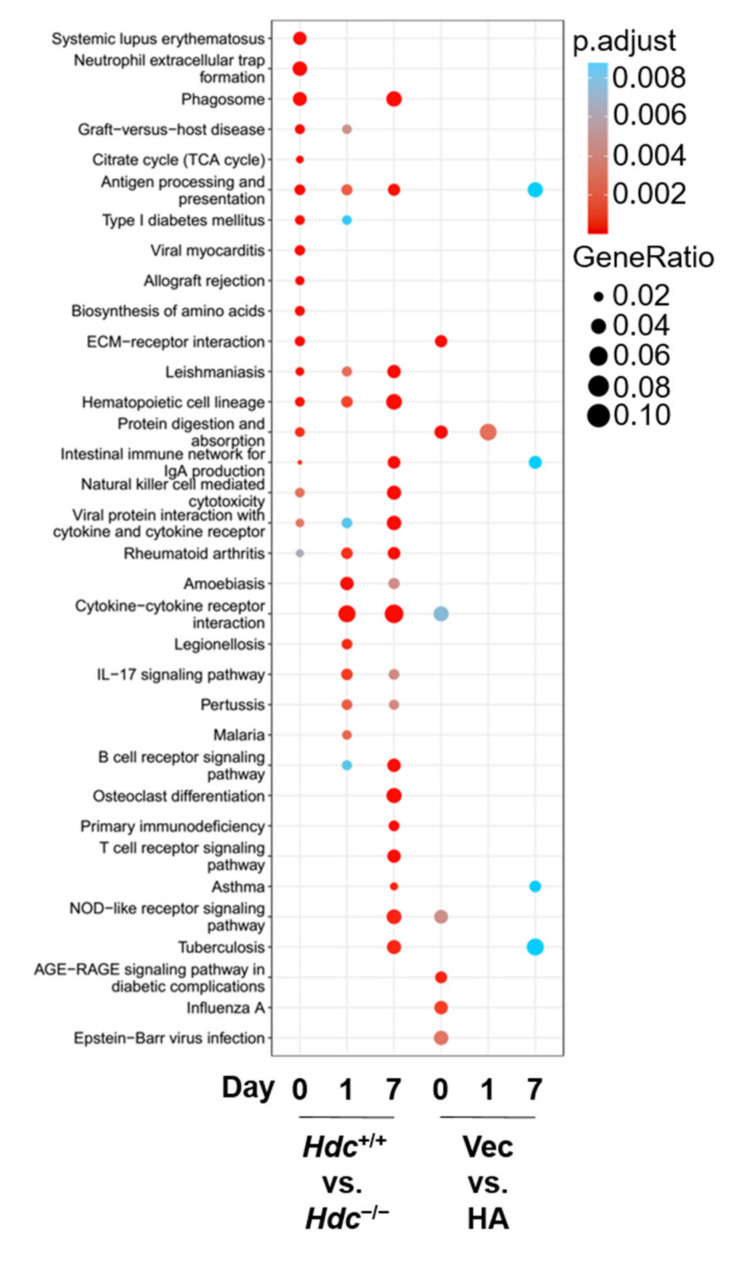
KEGG analysis of differentially expressed genes. The size of the point represents the ratio of gene numbers enriched in a given pathway in the background, whereas the background of the gene at each time point is the input of all the gene sets at that time point. The color of the dots represents the adjusted *p* value, whereas the blue to red gradient shows the statistical significance from low to high values.

**Figure 5 ijms-22-07401-f005:**
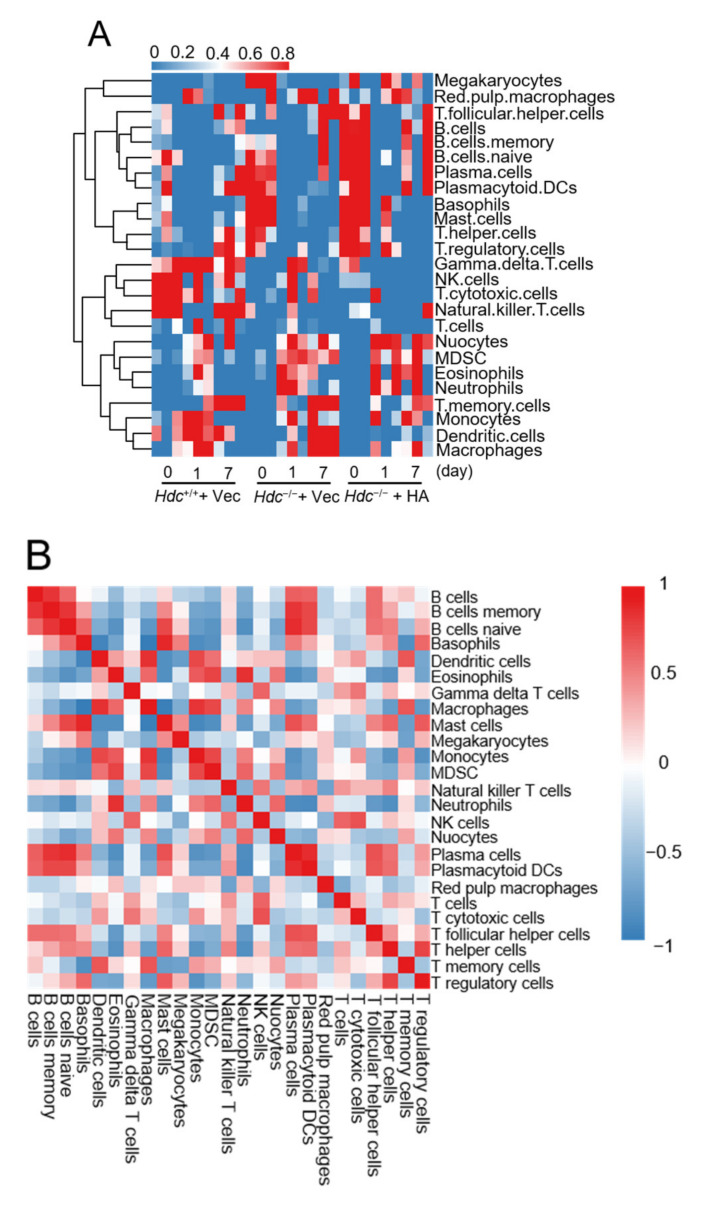
Infiltrated immunocytes in hearts from mice with different treatments. (**A**) Heatmap of the immunocytes in each group. (**B**) Correlation of the heatmaps of all the immunocytes. The color of the small squares represents Pearson’s correlation coefficient between the two immunocytes on the horizontal and vertical coordinates, and the blue to red gradient shows the coefficients from low to high values. (C and D) Summary of the estimated immune scores (**C**) and estimated fractions (**D**) of 22 subtypes of immunocytes from the ssGSEA algorithm in all groups. Each bar chart exhibits the cell proportions of each samples and the various colors represent the 22 immunocytes with annotations beside the legend.

**Figure 6 ijms-22-07401-f006:**
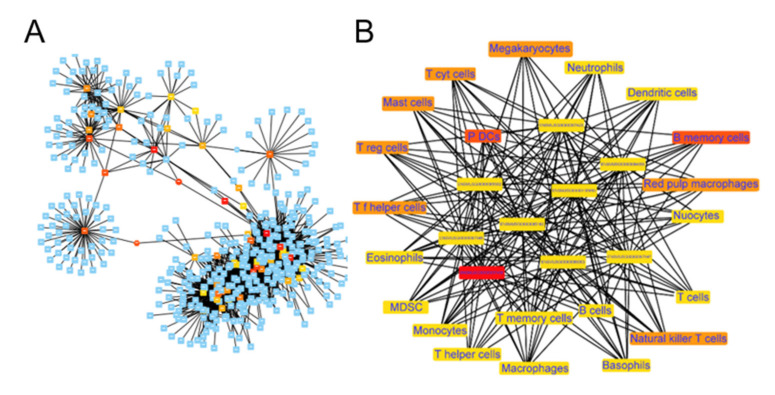
LncRNA–mRNA–immunocyte correlation network. (**A**) The co-expression network shows the correlation between immune-related lncRNAs and differentially expressed mRNAs, and the cutoff value was set with a threshold of correlation > 0.9 and *p* adjusted < 0.01. The yellow, orange, and red nodes represent lncRNAs by category according to the number of their connections with other genes, and blue nodes represent mRNAs. (**B**) The co-expression network of lncRNAs and infiltrated immunocytes. 35 key nodes are shown in the network. (**C**) The Sankey diagram summarized the relations between lncRNAs, the types of immunocytes, and the GO terms of correlated protein-coding mRNAs (pc-mRNA).

**Figure 7 ijms-22-07401-f007:**
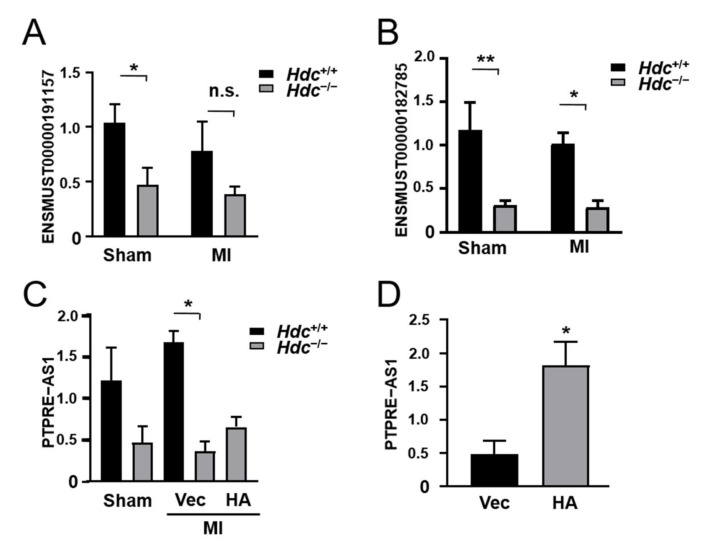
Verification of immune-related lncRNA levels by qRT-PCR. (**A**–**C**) qPCR results of immune-related lncRNA expression in *Hdc*^+/^^+^ and *Hdc*^−/−^ mice post MI operation. (**D**) The expression of PTPRE-AS1 in RAW264.7 cells with histamine treatment. n = 3–5, * *p* < 0.05, ** *p* < 0.01, n.s. no significance.

**Table 1 ijms-22-07401-t001:** Quantitative real time PCR Primers.

Primer	Forward (5′-3′)	Reverse (5′-3′)
ENSMUST00000180693 (PTPRE-AS1)	CAGTGAATGAGTGTGGCTCCTG	ACATGTAGAGTGTCCCTCGTTG
ENSMUST00000191157	CTGCCGCTAAGAAGGCGATT	CCCCTGCCAACCCATTCTT
ENSMUST00000182785	TGGTGGTTCCTGACTGTGGAC	AGCACCCACCCAAACAAGTCT
*Actb*	GGCTGTATTCCCCTCCATCG	CCAGTTGGTAACAATGCCATGT

## Data Availability

The detailed data used to support the findings of this study are available from the corresponding author upon written request. The RNA sequencing data are available in https://bigd.big.ac.cn/ under BioProject number PRJCA004459.

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
