# Peer review of "Analysis of Immune Associated Co-Expression Networks Reveals Immune-Related Long Non-Coding RNAs during MI in the Presence and Absence of HDC"

_ijms, 2021, doi:10.3390/ijms22147401_

Round 1
Reviewer 1 Report
Dear Authors,
The manuscript entitled "Analysis of immune associated co-expression networks reveals immune-related Long Non-coding RNAs during MI in the presence and absence of HDC" approaches the importance of Histidine Decarboxylase and immune related-lncRNA in gene expression during the progression of MI. It is very interesting and brings great contribution to the area. The manuscript is well written and has flow reding. However, some suggestions are below:
- Abstract:
1) Missing conclusion
- Results:
1) Figure 1A has an error on labeling. It should be Hdc-/- + HA instead of Hdc+/+ + HA. The deficient mice were treated only. Please correct it.
2) Be clearer on how clustering the Hdc groups. (line 120)
3) The authors describe “..The yellow and orange nodes represents lncRNA, blue nodes represent mRNAs.” In fig 6. It’s missing information about other colors and the blue ENSMUST00000191157 is later classified as lncRNA (line 237). Please clarify the misinformation.
4) Is the PTPRE-AS1 elevated in Hdc-/- + HA mice as well as shown in vitro?
5) References for “Activation, rather than inhibition 261 of histamine signaling, contributes to the recovery of MI or heart failure” (line 261)
Methods:
1) Have the mice really been housed at 22 ℃? It’s cold to them and interfere with data collection. See ref: Keijer J, Li M, Speakman JR. What is the best housing temperature to translate mouse experiments to humans? Mol Metab. 2019 Jul;25:168-176. doi: 10.1016/j.molmet.2019.04.001. Epub 2019 Apr 6. PMID: 31003945; PMCID: PMC6599456.
Conclusion:
1) First sentence is extremely long and need to be reformulated.
2) Conclusion should be mor clear, showing what the data suggest
Author Response
Deeply thanks for the positive comment. All the comments are valuable. We have performed some new experiments and made some revision to detailly address the reviewer’s concerns each by each.
- Abstract:
1) Missing conclusion
Response: Thanks for the advice. We have added brief conclusion at the end of Abstract as shown on Line 28 to Line 33.
- Results:
1) Figure 1A has an error on labeling. It should be Hdc-/- + HA instead of Hdc+/+ + HA. The deficient mice were treated only. Please correct it.
Response: We are really sorry for the mistake. We have corrected it in the revised version. And we also have checked the other labels in all the figures carefully again.
2) Be clearer on how clustering the Hdc groups. (line 120)
Response: This is a good advice. The clustering of differentially expressed genes regulated by Hdc/histamine is based on the different trends and different degrees of the alteration of their levels during the progress of MI. We have made it clear in the revised version as shown on Line 154 and Line 159.
3) The authors describe “The yellow and orange nodes represents lncRNA, blue nodes represent mRNAs.” In fig 6. It’s missing information about other colors and the blue ENSMUST00000191157 is later classified as lncRNA (line 237). Please clarify the misinformation.
Response: Thanks for the advices. 1) There are four different color nodes including yellow, orange, red and blue in total in Fig.6A. Beside mRNAs are labeled as blue nodes, lncRNAs are indicated as yellow, orange and red nodes by category according to the number of their connections with other genes. We had made it clear in the legends of Figure 6A as shown from Line 246 to Line 251. 2) In Fig.6C, different colors are used to distinguish different lncRNAs in the second column and their correlated immunocytes and GO terms. Blue is not used to distinguish mRNA.
4) Is the PTPRE-AS1 elevated in Hdc-/- + HA mice as well as shown in vitro?
Response: This is a great advice. According to the renewed Fig.7C, histamine treatment elevated the PTPRE-AS1 level reduced by Hdc deficiency, but which didn’t reach the significance. We speculated that this may be because PTPRE-AS1 is not specifically expressed in immune cells, so the difference of its expression in immune cells regulated by Hdc/histamine would be masked by its expression in other cells such as cardiomyocytes and cardiac fibroblasts. Thus, we explored the effect of histamine on the expression of PTPRE-AS1 in vitro as shown in Fig.7D. We had explained this concern in the revised manuscript as shown on Line 277 to Line 286.
5) References for “Activation, rather than inhibition 261 of histamine signaling, contributes to the recovery of MI or heart failure” (line 261)
Response: Thanks for the important reminding. We had added “Noguchi, K., et al., Histamine receptor agonist alleviates severe cardiorenal damages by eliciting anti-inflammatory programming. Proc Natl Acad Sci U S A, 2020. 117(6): p. 3150-3156” as References on Line 498.
Methods:
- Have the mice really been housed at 22 ℃? It’s cold to them and interfere with data collection. See ref: Keijer J, Li M, Speakman JR. What is the best housing temperature to translate mouse experiments to humans? Mol Metab. 2019 Jul;25:168-176. doi: 10.1016/j.molmet.2019.04.001. Epub 2019 Apr 6. PMID: 31003945; PMCID: PMC6599456.
Response: Greatly thanks for broadening our knowledge. We have carefully studied the reference recommended by the reviewer as above. With rigorous experimental demonstration, it newly recommended that the temperature range from 25.5 °C to 27.6 °C is most suitable for housing C57BL/6 mice to mimic human thermal relations. In our work, Balb/c mice were housed at 22 ℃ under a 12 h light/dark cycle, which is controlled by the animal management system according to the previously recommended temperature and can't be adjusted by individuals. We had added this concern in Method part to remind our readers as shown from Line 397 to Line 399.
Conclusion:
- First sentence is extremely long and need to be reformulated.
Response: Thanks for the advice. We have reformulated it as shown from Line 459 to Line 460.
2) Conclusion should be more clear, showing what the data suggest
Response: Thanks for the advice. We have reorganized it as shown from Line 460 to Line 463.
Reviewer 2 Report
The manuscript by Zhang et al. points to an interesting mechanism of the intrinsic linkage between ischemic injury and immune responses, involving immune‐associated lncRNAs and their roles in immune modulation after MI. Further, they demonstrated Hdc/histamine signal participated in regulating the pattern and the correlation of the immunocytes infiltrated into heart post MI at least partly via affecting the levels of imm‐lncRNAs. Although these findings are interesting, consideration of the following points should further strengthen and improve the overall presentation.
- Histological quantification of myocardial infarct should be included.
- In addition to ejection fraction, the authors should show other key echocardiographic parameters (e.g., systolic and diastolic volumes).
- Why was the RAW264.7 mouse macrophage cell line used for in vitro study? It was mentioned that ENSMUST00000181565 and ENSMUST00000180748 were highly correlated with macrophages. However, in immune cell profiles, there is no information about macrophages.
- In Figure 1A, the label of Hdc+/+ +HA is not correct.
- Some terms (e.g., MDSCs, CANTOS, and COLCOT ) should be present as the full name when they are first shown in the article.
Author Response
We appreciate the positive comment very much. We have performed some new experiments and made some revision to address the reviewer’s concerns as detailed in each specific point.
- Histological quantification of myocardial infarct should be included.
Response: This is an excellent suggestion. Actually, we had performed Masson's trichrome staining of cardiac sections 7 days post MI to determine the effect of Hdc/histamine on cardiac infarct size in HDC-/- mice. The results are shown as in Figure 1E with the corresponding text from Line 78 to Line 83 in the revised manuscript.
- In addition to ejection fraction, the authors should show other key echocardiographic parameters (e.g., systolic and diastolic volumes).
Response: This is a good advice. The important echocardiographic parameters including End Diastolic Volume and End Systolic Volume were showed in Figure1C and 1D with the corresponding text from Line 75 to Line 78 in the revised manuscript.
- Why was the RAW264.7 mouse macrophage cell line used for in vitro study? It was mentioned that ENSMUST00000181565 and ENSMUST00000180748 were highly correlated with macrophages. However, in immune cell profiles, there is no information about macrophages.
Response: Thanks for the advice. The information of macrophages in immune cell profiles had been included in Figure 5A, 5B, 5C and 5D. Maybe we didn't have a key description in the text that you didn’t notice it. In the revised version, we describe it more with the corresponding text from Line 198 to Line 200.
- In Figure 1A, the label of Hdc+/++HA is not correct.
Response: We are really sorry for the mistake. We have corrected it in the revised version. And we also have checked the other labels in all the figures carefully again.
- Some terms (e.g., MDSCs, CANTOS, and COLCOT) should be present as the full name when they are first shown in the article.
Response: Thanks for the important reminding. We have presented the full name of all abbreviations including MDSCs, CANTOS, and COLCOT when they first appeared in the revised version with the corresponding text from Line 195 to Line 196 and Line 300 to Line 302.

Round 2
Reviewer 2 Report
The authors have added more data and modified the manuscript to address comments from previous reviews.
However, there is no information of Masson's trichrome staining in the Methods.
Author Response
Thanks for the reminding very much. We have added the method of histological analysis on Line 414 to 417.
